# Checkpoint Inhibitors and the Gut

**DOI:** 10.3390/jcm11030824

**Published:** 2022-02-04

**Authors:** Tuan Tran, Nguyen Giang Tien Tran, Vincent Ho

**Affiliations:** School of Medicine, Western Sydney University, Campbelltown, NSW 2560, Australia; trannguyengiangtien1998@gmail.com (N.G.T.T.); V.Ho@westernsydney.edu.au (V.H.)

**Keywords:** inflammatory bowel, oncology, PDL-1, CTLA-4, cancer, checkpoint inhibitor, immunology, microbiome

## Abstract

Checkpoint inhibitors have revolutionized treatments in modern oncology, including many conditions previously relegated to palliative therapies only. However, emerging recognition of checkpoint inhibitor-related adverse events has complicated the status of checkpoint inhibitor-related therapies. This review article discusses gastrointestinal adverse events as a result of checkpoint inhibitor therapy, as well as limitations of current guidelines, thus providing recommendations for guideline revision and future study direction.

## 1. Introduction

Immune checkpoint inhibitors (ICI) have revolutionized modern oncological treatment by targeting the tumor’s ability to evade cytotoxic T-cells’ surveillance, thus improving both survival rates and disease-free condition, particularly in melanoma treatment [1]. A recent meta-analysis by Khan et al. [2] demonstrated ICI superiority compared to chemotherapy when treating non-small cell lung carcinoma. This was corroborated by a different systematic review showing significantly higher overall survival of combining chemotherapy with ICI or ICI monotherapy (specifically, atezolizumab or pembrolizumab) for advanced non-small cell lung carcinoma compared to chemotherapy alone [3]. A meta-analysis by Lehrer et al. [4] suggested that using ICI in combination with brain surgery could potentially improve safety and increase overall survival rate, although the author cautioned that a firm recommendation would need further prospective studies. Immunotherapy, compared to conventional cancer therapy (traditional chemotherapy, radiotherapy, and surgery), has better efficacy for rapidly proliferating cancer because it curbs the tumor’s rapid division and ability to evade immune surveillance [5]. Nonetheless, emerging evidence demonstrates that ICI is associated with significant side effects. In the gastrointestinal tract (GIT), ICI therapies often result in the development of esophagitis, gastroenteritis, and colitis. ICI-related GIT events, which account for 30% of all ICI-related adverse events, occur in 0.3–7% of the treatment population group, thus making it a highly significant issue [6]. Diagnosing and treating immune-related adverse events in the GIT is a challenging issue for physicians worldwide. We review the current literature pertaining to ICI-related adverse GIT events, their postulated pathophysiology, endoscopic features, and clinical management.

## 2. Epidemiology

Immune-related adverse events (irAEs) are reported in up to two-thirds of all patients undergoing immunotherapy. Amongst this group, up to one-third develop GIT symptoms, most commonly diarrhea. A heterogenous group of pathological conditions including ulcers, esophagitis, gastritis, and enterocolitis [7] are seen in association with ICI therapies. Amongst these, colitis is most well-studied and reported.

## 3. Risk Factors

Risk factors for development of irAEs in the gastrointestinal tract include gut microbiome, pre-existing autoimmune diseases, previous irAEs, and chronic use of anti-inflammatory agents [6]. A systematic review found that colitis more commonly occurs with CTLA-4 inhibitors compared to PDL-1 inhibitors (OR:8.7, 95%CI: 5.8.12.9) [8].

Gut microbiota composition has been shown to be a strong predictor of ICI-induced colitis. Dubin et al.’s [9] prospective study demonstrated that patients who remained colitis-free after ipilimumab treatment had a higher baseline level of Bacteroidetes in their gut microbiome composition compared to those who developed colitis. This was corroborated by Chaput et al. [10], who also found high levels of Bacteroidetes in patients who did not develop immune-related colitis, but high levels of Firmicutes in severe irAE colitis. Despite their small-scale (*n* < 100), these two studies provide the basic foundation for understanding the relationship between gut microbiome and immune-related colitis. Given the complexity of the human gastrointestinal microbiome, it is of considerable interest to elucidate definitively the relationship between microbiome and immune-mediated colitis.

Another risk factor for developing irAEs is the presence of autoimmune disorders such as systemic lupus erythematosus or rheumatoid arthritis, since these conditions often flare-up during ICI treatment. However, these exacerbations are typically mild and can usually be managed without need for treatment termination [11].

Of some importance, patients who have previously developed an immune-related adverse event to therapy with one class of ICI are at increased risk of developing further adverse events to a different class of ICI [12].

In contrast to other known autoimmune diseases, inflammatory bowel disease is not known to be a risk factor for irAEs [6]. This is a controversial statement as having inflammatory bowel disease is usually an exclusion criterion for most ICI trials [13]. Although large-scale studies have demonstrated the occurrence of similar pathophysiological processes in immune-mediated colitis and inflammatory bowel disease, no studies have identified a causal relationship between these two conditions [6,14]. However, it is recognized that a significant proportion of patients with known stable inflammatory bowel disease do experience colitis relapses during ICI treatment [15].

Furthermore, another controversial issue is gut toxicity from previous chemotherapy and radiotherapy, and their effect on the risk profile for ICI-induced colitis. Studies carried out on lung cancer patients who received ICI with previous chemotherapy and radiotherapy did not show an increased risk of ICI colitis [16]. However, the sample sizes of these studies were small. More studies with larger sample size are required before the relationship between chemotherapy/radiotherapy and ICI-induced colitis can be better defined. 

Chronic use of anti-inflammatory agents such NSAIDs has been shown to be associated with a higher risk of developing irAEs. A series of case studies by Marthey et al. [17] identified a correlation between chronic NSAID use and increased risk of ipilimumab-induced enterocolitis.

Khoja et al. [8] suggested that tumor histology could be predictive of irAE frequency/melanoma patients undergoing ICI treatment have a higher frequency of gastrointestinal irAEs compared to non-small cell lung carcinoma or renal cell carcinoma patients undergoing the same treatment. These findings were replicated in a recent single-center retrospective case series [18]. However, such interpretations should be made with caution given that an overwhelming majority of patients in these studies had melanoma. A meta-analysis confirmed that, for PDL-1 therapies, the incidence of colitis appeared to be higher in patients with melanoma, white race, and stage III–IV cancer [19]. 

## 4. Pathophysiology

### 4.1. Role of Checkpoint Inhibitors

Tumors evade the adaptive immune system by manipulating immune tolerance and immune resistance mechanisms. Tumors typically express tumors-associated antigens, which are picked up by antigen-presenting cells. Antigen-presenting cells activate T-cells through interactions with multiple T-cell receptors (TCR), one of which, CTLA-4 (CD152), is a negative regulator. The activated T-cells then target and destroy tumor cells. CTLA-4 activity at the tumor site is downregulated by regulatory T-cells in a process known as immune tolerance. This process is activated through specific receptors such as programmed cell death protein 1 (PD-1), otherwise known as CD279 [20]. CTLA-4 receptor on T-cell surface mediates its inhibition by competitively binding to B7 protein on antigen-presenting cells. This prevents CD28—which is another B7 ligand—from binding to B7 protein. This negative costimulatory signal blunts T-cell activation and responsiveness [21]. By artificially blocking the binding of CTLA-4 to B7 but still preserving CD28 binding, this allows the promotion of T-cell activation and proliferation [22]. Ipilimumab is a humanized monoclonal antibody that binds to CTLA-4, thus preventing it from binding to B7 protein without blocking CD28 signaling via B7. The blockade of CTLA-4 signaling promotes T-cell activation and proliferation, henceforth contributing to a T-cell-mediated immune response against tumor cells [22,23]. However, CTLA-4 blockade also concomitantly inhibits T-cells’ regulatory function. This results in excessive T-cell proliferation, hence inducing dysregulation of gastrointestinal mucosal immunity and eventually the development of immune-related adverse events, most evidenced in the lamina propria and the submucosa along the gastrointestinal tract [24].

Another crucial checkpoint in the human T-cell response against tumors is PD-1, which is expressed on the surface of activated T-cells. Interactions between PD-1 (on activated T-cells) and programmed cell death ligand 1 (PDL-1 on antigen-presenting cells leads to T-cell inactivation and apoptosis, hence the name ‘programmed death ligand’) [25]. This is a basic mechanism of immune tolerance, especially in peripheral tissues. Human monoclonal IgG4 antibodies against PD-1, such as nivolumab and pembrolizumab, and human monoclonal IgG1 against PDL-1, such as avelumab, competitively bind to these receptors. The anti-PD-1 and anti-PDL-1 antibodies exhibit synergistic activity in preventing CTLA-4-mediated downregulation, reducing T-cell apoptosis and increasing the net activity of activated T-cells against tumor cells. However, in an area where there is a strong presence of immune cells, such as the gastrointestinal tract, persistence of activated T-cells can lead to damage to healthy tissues and development of irAEs [20]. 

Please see Figure 1 for simplistic model outlining CTLA-4 and PDL-1 interaction.

### 4.2. Pathophysiology of Immune-Related GIT Events

There are several postulates concerning the pathophysiology of immune-related GIT events.

One theory is that ICI removes protection from autoimmunity [26]. In essence, it is postulated that ICI reduces T-cells’ self-downregulatory mechanisms, thus promoting T-cell proliferation, as in the case of CTLA-4 inhibitors. In addition, ICI such as inhibitors of PD-1 and PDL-1 promote activity of already activated T-cells [20]. This combination of increasing T-cell proliferation and upregulation of T-cells leads to high-level secretion of pro-inflammatory CD4 T-helper cell cytokines and cytotoxic CD8 T-killer cell tissue infiltration, which are well-demonstrated in ICI-related colitis [27].

Another theory is that ICI induces a reduction in regulatory T-cell activation and proliferation [28]. Regulatory T-cells naturally have higher expression of CTLA-4 receptors, and thus are more at risk of becoming inactivated by humanized monoclonal antibody [29]. The reduction in T-regs leads to an inflammatory state not dissimilar from graft-versus-host disease [30]. This involves uncontrolled T-cell and B-cell proliferation in multiple organs, leading to a widespread multi-organ inflammatory condition, typically involving dermatitis, nephritis, hepatitis, and colitis.

ICI also interacts with interleukin-17 production [31]. Mice studies showed that the administration of PDL-1 antibodies can dampen IL-17 concentration in the colon, which may explain why PDL-1 therapies have fewer colonic irAEs compared to CTLA-4 [31,32,33]. Changes in the population of resident CD8+ T-cells in the colon were also speculated to enhance inflammatory response [34].

An emerging theory has postulated a crucial role by gut microbiota in the pathophysiology of gastrointestinal irAEs. Studies by Dubin et al. [9] show a specific enrichment of Bacteroidetes in colitis-resistant patients. Bacteroidetes comprise a major, albeit largely understudied, phylum in the human GIT microbiome. It is postulated that Bacteroides members may limit inflammation by stimulating T-regulatory cell differentiation. Bacteroides have been shown to produce riboflavin metabolites which could potentially activate innate T-regulatory cells within the mucosa, thus promoting immune tolerance in the GIT [9]. DNA sequence analysis of gut Bacteroides species, and molecular and biochemical analysis of riboflavin metabolism, which may cause activation of T-regs, are subjects of current research.

Clinically, the presentation of ICI-mediated gastrointestinal complications is highly heterogenous. Currently, it is not clinically feasible to distinguish whether gastrointestinal complications were due to pre-existing inflammatory bowel disease flare-up or induced by ICI therapy [35]. Patients with existing inflammatory bowel disease undergoing ICI treatment remain vulnerable to life-threatening gastrointestinal complication [32]. The most recent systematic reviews and meta-analyses were underpowered to determine whether patients with pre-existing autoimmune condition would benefit or be worse off if they continue with ICI therapies, as many of these patients were excluded from clinical trials [13]. Hence, without explicit and evidence-based guidance, managing physicians need vigilant monitoring for signs and symptoms of toxicities.

The sections below describe the clinical presentation of ICI-induced irAEs in different parts of the gastrointestinal tracts.

## 5. Oral Cavity

The oral cavity is well-known to be affected by immune modulators, which cause a variety of pathologies of wide-ranging severity. Unfortunately, oral disorders are under-reported, as systemic signs are usually only seen in rare cases.

Cutaneous lichenoid reaction is the most common ICI event in the oral cavity, having an incidence of up to 17% [36]. Histopathologic findings are largely non-specific, typically involving mucositis with dense lymphocytic and eosinophilic infiltration and small amounts of clefting without fulminant ulcer [37]. Immunofluorescence analysis shows similar presence of CD4 and CD8 cells and negative for immunoglobulin at the basement membrane [38,39].

Apart from superficial lichenoid lesions, 1% of all patients treated with immunomodulators develop superficial oral mucosal blisters, also known as mucous membrane pemphigoid [40]. Mucous membrane pemphigoid typically involves relatively deep infiltrates and histopathologic immunofluorescence findings that typically indicate subepithelial cleavage and linear immunoglobulin deposits at basement membrane [36].

Another lesion, erythema multiform major, has been associated with the use of PDL-1 antagonist and CTLA-4 antagonist [36]. These lesions are usually more serious and extensive than superficial blisters. Histopathologic findings are largely nonspecific, but considerable inflammatory infiltration of predominantly polymorphonuclear leucocytes is observed [41].

While lichenoid, membrane pemphigoid, and erythema multiforme major are largely local reactions, systemic inflammatory responses are not uncommon in severe cases. Stevens–Johnson syndrome-like reactions, toxic epidermal necrolysis-like reactions, and Sjogren syndrome are amongst oral cavity immune modulator-induced adverse events with systemic features. The presence of these conditions is usually an indication to discontinue immunomodulators [36,42].

Stevens–Johnson syndrome and toxic epidermal necrolysis are well-feared complications of immunomodulators. Although rare (<1% incidence among patients treated with ICI therapy), they are usually associated with PDL-1 and PD-1 antagonist treatment [36]. Lesions are usually not limited to the oral cavity, and are characterized by widespread dusky macules or flat, atypical targetoid lesions involving a high index of body surface area, usually with signs of systemic inflammatory response syndrome [43].

Finally, Sjogren syndrome is a well-documented ICI adverse event with clinical features typically including fulminant multi-gland destruction in the oral cavity. Histopathological features of gland biopsies typically show extensive sialadenitis with T-cell infiltration (more CD8 than CD4) [42].

## 6. Esophagus

Esophageal involvement in immunomodulator-mediated adverse events is rare, with only two documented case reports to date [44,45]. Patients with ICI-induced esophagitis typically present with erythematous superficial erosions and diffuse mucosal nodularity that are highly atypical compared to other forms of esophagitis. Unfortunately, reports of direct immunofluorescence analyses of these two cases were limited. Esophageal ulceration and stenosis have also been reported [45].

## 7. Stomach and Small Bowel

Adverse events affecting the stomach have been well-documented following immunomodulator therapy. Symptoms typically include: abdominal pain, gastroesophageal reflux, and even diarrhea [46]. Multiple mid-size studies have shown that gastric irAEs exhibit patterns of chronic inflammation, similar to chronic gastritis [46]. Specifically, in the stomach, ICI-associated gastritis is characterized by mild lamina propria inflammation associated with relatively few lamina propria-located CD20 B-cells, and by severe intra-epithelial lymphocytic infiltration associated with large numbers of intraepithelial CD8 T-cells. Fewer lymphoid aggregates are associated with IM gastritis compared with H. pylori gastritis. Aside from a higher incidence of gastric lymphoid aggregates in patients on anti-PD-1 therapy than in patients on anti-PDL-1 therapy, no morphologic or immunohistochemical differences between the two drugs were found in gastric mucosal biopsies.

While it remains true that gastritis is a well-documented ICI-related GIT complication, there is a paucity of large-scale studies and thus a lack of formal standardization of disease grading and treatment guidelines. Small-scale studies by Johncilla et al. [47] and Irshaid et al. [46] show common themes of gastritis, including raised intra-epithelial lymphocytic infiltration and loss of duodenal crypts. These are largely underpowered studies, thus necessitates more large studies before a formal guidelines for grading and treatment of ICI-related gastritis could be made.

Within the duodenum, checkpoint inhibitor-associated mucosal injury is characterized by marked villous blunting and increases in infiltrating intraepithelial lymphocytes. The main histologic discriminator is the presence of activity (defined as neutrophilic infiltrates and/or erosions), which is found in 100% of ICI-associated duodenitis biopsies. However, because acute inflammation can occasionally be seen in celiac disease, the morphologic similarities between the two conditions can potentially be diagnostically misleading. Further, it has been established that ICI therapy can induce manifestations of celiac disease in genetically at-risk individuals, as described in a recent case report of diet-responsive celiac disease developing after anti-CTLA-4 therapy [48]. In addition, ICI injury was postulated to exacerbate the immune response to H. pylori infection in a patient receiving anti-CTLA-4 antibody, leading to severe hemorrhagic gastritis [46].

Despite being common disease occurrences, checkpoint-related duodenitis and gastritis are not well-studied. International guidelines do not yet exist for grading disease severity. In addition, treatment guidelines do not yet exist, forcing clinicians to depend on guidelines established for treating ICI-related colitis. With the unique involvement of possible longer GIT segments and unique features of the small bowel, there is a need to improve current guidelines’ comprehensiveness more comprehensive guidelines in management of checkpoint-related duodenitis and gastritis.

## 8. Colon

Up to 30% of all ICI-induced GIT events involve the colon [6], making this site the most affected by checkpoint inhibitor GIT-related adverse events.

Immune checkpoint inhibitor-related adverse events are heterogeneous, with five known grades according to current international guidelines [6]. Although variable, histopathologic findings typically involve epithelial infiltration by neutrophils and macrophages with crypt formation or even crypt abscesses [14]. Colonoscopic biopsy has a 90% sensitivity for detecting grade I (mild) colitis, which has features of microscopic inflammation [49]. However, it is postulated that mild colitis is an over-diagnosed phenomenon, as 37% of patients with clinical diagnosis of immune-mediated colitis have a normal colonoscopy, and 5.6% of patients of all patients diagnosed with ICI colitis also have normal biopsies [14]. It is questionable that, with negative endoscopies, and negative biopsies, subjective symptoms such as abdominal discomfort (including post-prandial bloating and early satiety) and self-reported stool frequencies of these symptoms could be reliably used to diagnose ICI colitis. Interestingly, despite new research directions that emphasize histopathology, current guidelines do not suggest grading based on histopathological findings from endoscopy.

One unique feature of ICI colitis is that histologic changes usually precede the onset of diarrhea or colitis, again emphasizing the specificity of histopathologic sampling in IMC diagnosis. Coutzac et al. [14] showed that inflammatory changes preceded onset of symptoms by 3 weeks. Biopsies performed prior to onset of symptoms in ipilimumab-treated patients identified a group of ‘pre-symptomatic patients’ with a distinct neutrophilic infiltrate and cryptitis. Most patients in this group developed symptoms later, with a correlation between histopathological findings and diseases severity such as crypt abscesses, as well as glandular destruction and mucosal erosions.

In contrast to inflammatory bowel diseases, early studies on ICI-related colitis failed to identify features of chronic inflammation such as crypt architectural distortion, basal plasmacytosis, presence of granulomas, Paneth cell metaplasia, or pyloric metaplasia [6,14]. However, some recent studies have shown histopathological evidence of chronic inflammation in ICI colitis arising months after onset of symptoms [50].

Thus, there is an emerging paradigm shift that recognizes the similarity between IMC and IBD. Both conditions are recognized as having similar chronic pathology with similar predisposal to flares with certain triggers even when original insult was removed. One recent study compared immunohistochemistry and flow cytometry of colonic biopsies taken from patients with IMC (further divided to PD-1-treated and CTLA-4-treated) or with IBD. While both conditions showed similar histopathologic features of leucocytic infiltration of lamina propria, remarkably, IBD- or anti-PD-1-induced colitis shared similar features of predominant regulatory T-cells in the lamina propria, whereas in anti-CTLA-4-induced colitis, CD8 T-cells were more common [14]. This leads to the question: should ICI colitis be treated and perhaps graded using pre-existing IBD guidelines?

While most studies grade IMC based on symptoms, a few have suggested the use of histopathology to grade the disease. Two common scoring systems are the Mayo score and the van der Heide score [51]. These histopathology scoring systems have been used since the 1980s to assess disease severity. In view of our new understanding of histopathologic and immunologic complexities of IMC, it will be important to develop new grading tools.

## 9. Motility

Despite extensive histopathological research in different parts of the GIT, gastrointestinal motility is not fully understood in immune checkpoint-mediated adverse events. There is a single documented case demonstrating dysmotility resulting from Immune Checkpoint Inhibitor therapy [52]. Most current research has focused on either the oral cavity or the colon, as the enteric plexus of the small bowel remains poorly understood. As the use of checkpoint inhibitors becomes more mainstream, there will be more complex GIT-related events recognized, which will drive research aimed at understanding the effects of ICI therapy on gut motility.

## 10. Treatment

Treatments for ICI-related adverse events largely consist of terminating the use of the offending agents, providing supportive care such as intravenous rehydration and correction of electrolytes, and, in severe cases, use of corticosteroid and anti-TNF-alpha [6]. These are similar to guidelines for treatment of ICI colitis from the Society for Immunotherapy of Cancer, the American Society of Clinical Oncology, and the European Society for Medical Oncology (Table 1).

There is an increased clinical use of vedolizumab, a humanized murine antibody with activity against the α4β7-integrin heterodimer on the surface of CD4+ T-cells, to treat ICI-related colitis [53]. It is understood that the α4β7-integrin binds to its ligand MAdCAM-1 on the endothelial surface of venules within the gut, thus making vedolizumab more gut-specific compared to traditional TNF-α antagonists [54]. Small-scale case reports have shown promising results in treating steroid-resistant and even infliximab-resistant ICI-related colitis [54,55]. Results are pending from a large randomized controlled trial, which is currently in its phase I/II [55].

The recommendations based on these guidelines include similar indications for, and dosing of, corticosteroids, and recommendations for further investigation. Some of the newer guidelines, as from the European Society for Medical Oncology, recommend initiating immunosuppression such as with Mycophenolate and TNF-alpha inhibitors as soon as infection and surgical pathology have been ruled out [6].

However, emerging findings suggest that these three main guidelines are not necessarily appropriate. For example, despite its common use as a mainstream treatment for ICI-related colitis, there is evidence that high-grade colitis is usually refractory to steroid treatments, and their use is associated with a high recurrence rate [51]. Their study identified risk factors for refractory response to steroid treatment including macroscopic ulcers, pancolitis, and high Mayo or van der Heide scores. Most patients with one or more of these risk factors were ultimately treated with infliximab, which produced an excellent response. The success of these largely small-scale case studies in identifying and treating steroid refractory ICI colitis has led to a more favorable view of treatment with TNF-alpha inhibitors and other immunosuppressive agents.

We note that no treatment guidelines yet exist for Immune Checkpoint-mediated adverse events in the oral cavity, esophagus, stomach, or small bowel. The guidelines included in Table 2 largely focus on the use of steroids and TNF-alpha antagonists. The guidelines from the different sources share a ‘shotgun approach’ of using blunt anti-inflammatory agents that have been found to work in other inflammatory bowel diseases. In view of the advances made in development of immunologically modified medications and a better understanding of the histopathology of immune checkpoint-mediated GIT adverse events, it is reasonable to include more-targeted treatments, and aim to improve current guidelines’ comprehensiveness by incorporating histopathology into the grading system. While traditional grading systems such as Mayo and van der Heide scoring have been useful, they are outdated. In detail, van der Heide score involves only a simplistic depiction of histopathology, namely, presence of ulcer, vascular patterns, and granularity [56]. Furthermore, Mayo score only provides one single scoring column for macroscopic mucosal appearance at endoscopy. Immunochemical staining and quantification of specific T-cell subsets are not included in these scores, but these metrics are now standard procedures for analysis of ICI colitis samples.

Ultimately, more studies that are specifically powered to stratify the treatment approach for different levels of severity of ICI-related GIT adverse events.

### 10.1. Blood Tests and Immunological Markers

The use of novel inflammatory markers in managing ICI colitis is relatively new. However, studies on ICI employing markers traditionally used for inflammatory bowel conditions, have produced controversial results. For example, as a marker for monitoring GIT inflammation, fecal calprotectin has excellent sensitivity and specificity, and it has proved useful in IBD monitoring. However, fecal calprotectin is not specific to ICI-induced colitis [6], and it is a poor predictor of disease time course and disease severity for ICI colitis [57]. Nonetheless, emerging evidence suggests that quantification of fecal calprotectin correlates with endoscopic severity in ICI colitis. These promising findings suggest that, in conjunction with additional prospective data, fecal calprotectin may find use as a non-invasive marker of histopathologic response to treatment or remission in ICI colitis [49].

The use of novel markers such as IL-17 has been considered [57] showed a specific correlation of elevated levels of IL-17 in post treatment patients who developed CTLA-4-induced colitis. This finding suggests that elevated IL-17 may have use as a risk factor for developing ICI colitis.

Another encouraging finding is that eosinophil levels may also have use as a predictor of risk for developing ICI colitis [58].

### 10.2. Guidelines for Diagnosis and Treatment

Current guidelines for management of GIT-related irAEs use symptomology to triage colitis grade (Table 2). None of the current guidelines use endoscopy to guide treatment although the American Society of Clinical Oncology recommends the use of endoscopy to diagnose concomitant opportunistic infection (Table 2). Furthermore, treatment triage depends solely on use of corticosteroid, electrolyte replacement, and possible TNF-alpha antagonist use. However, as previously mentioned, the blunt use of anti-inflammatory agents as symptomatic managements for irAEs is often complicated in high-graded lesions, thus predisposing patients to debilitating recurrences if ICI is not immediately ceased. Preventing recurrences, particularly in long-term treatment with ICI, is not yet adequately addressed by the current guidelines, and no reasonable solution other than terminating ICI treatment is proposed. There is a vague recommendation for endoscopy, but it is without clear direction to be used in response to specific histopathological phenomena, i.e., crypt formation [6].

All major guidelines recommend electrolyte optimization, appropriate rehydration, and timely withdrawal of immunotherapy. There are minor differences between recommendations for CTLA-4 inhibitor compared to PD-1 and PDL-1 inhibitors; the American Society of Clinical Oncology recommends early termination of CTLA-4 inhibitor, whereas other organizations recommend equal precautions regarding PDL-1 inhibitors and CTLA-4 inhibitors.

Interestingly, the major guidelines share similar views regarding early use of corticosteroids and similar dosing recommendations. However, recommendations for escalating to use of TNF-alpha antagonists are not clearly outlined. In view of the emergence of steroid-resistant colitis and the emphasis on early treatment with TNF-alpha antagonist based on histopathologic findings, new guidelines with emphasis on use of endoscopy to help guide treatment are necessary.

## 11. Future Directions and Recommendations

There is one conclusion that recent reviews on the topic of GIT-related irAEs share: this unique immunological phenomenon is becoming increasingly common and the guidelines are outdated.

The diagnosis of ICI-related gastroenteritis or colitis relies heavily on the clinician’s understanding and vigilant observations of the clinical presentation. There are no unique diagnostic criteria for upper gastrointestinal adverse events related to ICI. This is understandable, as upper gastrointestinal events such as gastritis, esophagitis, and enteritis are rare, and data largely come from small cross-sectional studies or case series. For the same reason, oral cavity-related adverse events are significantly under reported [36], further leading to fewer data that are collectable, and making future studies even more challenging. Consequently, there is a need for a new international consensus related to screening for irAEs in the upper gastrointestinal area.

In contrast, ICI-related colitis is well-studied, and guidelines are well-established for the early recognition of pathology. However, these guidelines are outdated when it comes to application to therapeutic approaches. The guidelines fail to incorporate the new findings pertaining to the presence of unique histopathological markers of disease severity. While Mayo score and van der Heide score have been highly successful in predicting prognosis and treatment response rates in other IBDs, their application to ICI colitis is not certain [51]. This is reflected, for example, in confusion among major guidelines regarding the circumstances under which TNF-alpha inhibitors should be introduced. There is a need for international consensus in relation to the incorporation of current immunologic and histopathologic findings to make the guidelines more reflective of current understanding of ICI colitis and to understand ICI colitis as an inflammatory bowel condition rather than an unpleasant side effect of oncological treatment.

Finally, a crucial new direction for GIT-related irAEs should be the early identification of highly susceptible patients based on microbiota. Currently, there are no feasible measures available for screening patients for ICI colitis risk [6]. However, there are emerging data from microbiome studies which indicate that levels of Bacteroidetes may be protective against colitis, while high levels of Firmicutes may be associated with susceptibility to colitis. These observations suggest the possibility of screening for high-risk patients, which, along with promoting vigilance toward ICI-related colitis, would provide an important tool for improving care of patients receiving checkpoint inhibitor-related therapies.

## Figures and Tables

**Figure 1 jcm-11-00824-f001:**
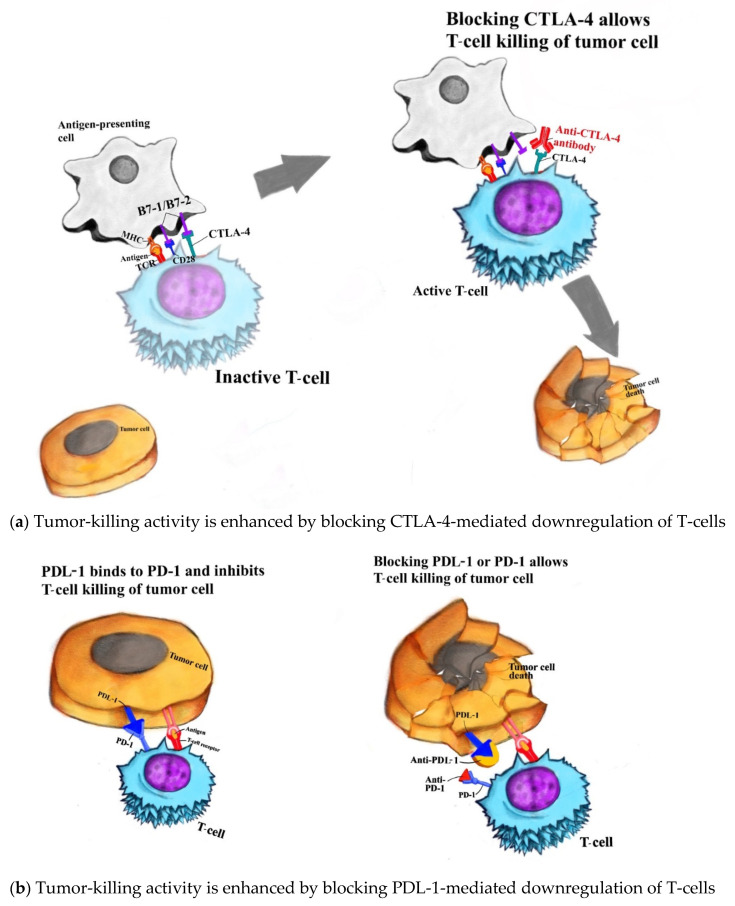
Immune checkpoint inhibitors (anti-CTLA-4 and anti-PDL-1 antibodies) enhancing tumor-killing activity (**a**) anti-CTLA-4 antibody’s tumor-killing activity; (**b**) anti-PDL-1 antibody’s tumor-killing activity.

**Table 1 jcm-11-00824-t001:** Treatment recommendations for different irAE-colitis grades. Reprinted with permission from Ref. [6]. Copyright 2019 Copyright Som et al.

Colitis Grade	Society for Immunotherapy of Cancer	American Society of Clinical Oncology	European Society for Medical Oncology
I	Continue Immunotherapy	Continue Immunotherapy	Continue Immunotherapy
II	Withhold immunotherapy Commence prednisone 1–2 mg/kg/day	Stop CTLA-4 inhibitor permanently Withhold immunotherapy Commence prednisone 1–2 mg/kg/day	Withhold immunotherapy If persists more than 3 days or worsens, treat with prednisolone 0.5–1 mg/kg/day Schedule colonoscopy but do not wait for colonoscopy to start therapy
III	Withhold immunotherapy Start intravenous prednisone 1–2 mg/kg/day Consider other anti-inflammatory agents, e.g., infliximab 5 mg/kg, or vedolizumab Consider endoscopy	Stop CTLA-4 inhibitor permanently Only consider restarting PDL-1 inhibitors if improved Consider prednisone 1–2 mg/kg per day If symptoms persist more than 3 days, may administer IV corticosteroid or infliximab Endoscopy only when patients may be at risk of opportunistic infections or consider starting infliximab	Withhold immunotherapy IV methylprednisolone 1–2 mg/kg/day If no improvement or worsening in 72 h, treat with infliximab 5 mg/kg (if no perforation, sepsis, TB, hepatitis, NYHA III/IV CHF) May consider other immunosuppressants: MMF 500–1000 mg BD or tacrolimus (plasma level aiming 10–15 ng/mL) Endoscopy prior to initiation of TNF-alpha inhibitors
IV	Cease immunotherapy indefinitely Same as grade III	Cease immunotherapy indefinitely IV corticosteroid until symptoms improve Early infliximab 5–10 mg/kg if symptoms are refractory to corticosteroid within 3 days	No recommendations regarding duration of immunotherapy cessation Same as grade III

**Table 2 jcm-11-00824-t002:** National Cancer Institute Common Terminology Criteria for Adverse Events guideline. Reprinted with permission from Ref. [6]. Copyright 2019 Copyright Som et al.

Grade of ICI-Related Colitis	Symptoms
I	Asymptomatic, less than 4 stools per day over baseline
II	Abdominal pain, mucus, blood in stool, more than 4–6 stools per day
III	Severe pain, fever, peritoneal signs, more than 7 stools per day
IV	Life-threatening consequences such as perforation, ischemia, necrosis, bleeding, toxic megacolon, hemodynamic collapse
V	DEATH

## Data Availability

Not applicable as this is a review article.

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
