# Peer review of "Checkpoint Inhibitors and the Gut"

_jcm, 2022, doi:10.3390/jcm11030824_

Round 1

Reviewer 1 Report

The subject of this article is the toxicity of Checkpoint inhibitors on the digestive tract.
The vast majority of articles describe all the toxicities of these molecules and when the subject is the gastroenterological toxicity, the major part of the article is devoted to hepatic toxicities.
This article is interesting because it deals with lesser-known toxicities (oral cavity, oesophagus, stomach, duodenum) and, of course, the better-known colic toxicity.
The article is clear, well written and well documented.

minor comment: the division of the paragraphs is curious with a paragraph 5, oral cavity, a paragraph 6  esophagus and then subparagraphs 6.1 stomach, 6.2 colon, 6.3 motility, 6.4 treatment.

major comment

the article does not address the practical questions that oncologists ask themselves on a daily basis
1) Can check point inhibitors be considered in patients with a history of Crohn's disease, rectocolitis, ulcers, hiatal hernia, etc.?
2) What is the additive risk of radiotherapy on the digestive toxicity of checkpoint inhibitors?
3) given the importance of checkpoint inhibitors in the therapeutic strategy of many cancers (melanoma, lung, liver, kidney, etc.), the question arises of reintroducing them after stopping checkpoint inhibitors for toxicity.

Nevertheless, as this seems to be a special issue, it is possible that the questions I am asking will be dealt with in another chapter.

Author Response

1st reviewer

The subject of this article is the toxicity of Checkpoint inhibitors on the digestive tract.
The vast majority of articles describe all the toxicities of these molecules and when the subject is the gastroenterological toxicity, the major part of the article is devoted to hepatic toxicities.
This article is interesting because it deals with lesser-known toxicities (oral cavity, oesophagus, stomach, duodenum) and, of course, the better-known colic toxicity.
The article is clear, well written and well documented.

✅ minor comment: the division of the paragraphs is curious with a paragraph 5, oral cavity, a paragraph 6 esophagus and then subparagraphs 6.1 stomach, 6.2 colon, 6.3 motility, 6.4 treatment.

Thank you for pointing out the minor errors. It should be 5 oral cavity, 6 oesophagus and then subparagraphs 7 stomach, 8 colon, 9 motility, and 10 treatment. This has been rectified.

major comment

the article does not address the practical questions that oncologists ask themselves on a daily basis
1) Can check point inhibitors be considered in patients with a history of Crohn's disease, rectocolitis, ulcers, hiatal hernia, etc.?

Thank you for this comment. Additional papers are now cited addressing the point above and mentioned in our manuscript.

2) What is the additive risk of radiotherapy on the digestive toxicity of checkpoint inhibitors?

Thank you for the comment.  From a common sense perspective, we would expect ICI to cause more digestive tract toxicity and with already a bowel already at risk, radiotherapy should be undertaken with caution. However, the data does show that the combination of chemotherapy and radiotherapy DID NOT increase the risk of ICI colitis (Tang et al, 2021).

3) given the importance of checkpoint inhibitors in the therapeutic strategy of many cancers (melanoma, lung, liver, kidney, etc.), the question arises of reintroducing them after stopping checkpoint inhibitors for toxicity.

Nevertheless, as this seems to be a special issue, it is possible that the questions I am asking will be dealt with in another chapter.

Thank you for the comment. The current meta-analysis cited was underpowered to answer this question but with further research this question could be answered. We recognise that it is likely another chapter will address this point in the special issue.

Reviewer 2 Report

This is an interesting review focusing on all aspects of GI irAEs induced by ICIs. The authors do a good job summarizing the available evidence. Nevertheless, there are significant issues that need to be addressed, in my opinion.

Line 16: I´m not sure ICIs enhance t-cell responses during oncogenesis. Conversely, ICIs enhance T-cell responses in established tumors > To date, there is no evidence that ICIs may act preventing tumors.

Line 18: I think the statement that ICIs have “better efficacy for rapidly-proliferating cancer” compared to other treatments (chemo, radiation, and surgery) is highly controversial. Plenty of tumors proliferate fast and do not benefit/minimally benefit from ICIs to date (e.g., small cell lung cancer and pancreas).

Risk factors: I would also include tumor type as a risk factor: different histologies lead to different toxicity profiles. Khoja L, Day D, Wei-Wu Chen T, Siu LL, Hansen AR. Tumour- and class-specific patterns of immune-related adverse events of immune checkpoint inhibitors: a systematic review. Ann Oncol. 2017 Oct 1;28(10):2377-2385. doi: 10.1093/annonc/mdx286. PMID: 28945858.

Pathophysiology: The mechanisms presented exemplify how the most well-known ICIs [anti-PD(L)1 and anti-CTLA4) work. However, why do irAEs occur in the gastrointestinal tract? What is unique about GI that leads to such a high % of AEs?

Lines 159-160: Neither Stevens-Johnson nor toxic epidermal necrolysis is “commonly associated” with anti-PD(L)1 treatments. These are rare events representing less than 1% of irAEs

6.1 Stomach and small bowel: I suggest the authors include a bit of the clinical presentation of such adverse events. How do these patients present? What are the typical side effects?

Line 223: I believe more clarity is needed in this paragraph. The authors affirm that colonoscopy has a high specificity (should be sensitivity??) in detecting mild colitis. Conversely, affirm that 37% of the clinical diagnosis of mild colitis have a normal colonoscopy, and the condition is over-diagnosed. I think these 2 passages are contradictory. In clinical practice, physicians often don´t use colonoscopy for diagnostic purposes. If the presentation is typical non-invasive tests such as CT may be performed, or even treatment may be initiated without diagnostic confirmation.  

6.6: I suggest the authors include some wording about Vedolizumab. This is a potentially less immune-suppressive agent in comparison to anti-TNF alpha drugs as it is more specific to the gut.

Author Response

2nd reviewer

This is an interesting review focusing on all aspects of GI irAEs induced by ICIs. The authors do a good job summarizing the available evidence. Nevertheless, there are significant issues that need to be addressed, in my opinion.

✅ Line 16: I´m not sure ICIs enhance t-cell responses during oncogenesis. Conversely, ICIs enhance T-cell responses in established tumors > To date, there is no evidence that ICIs may act preventing tumors.

Thank you for the comment. ICI indeed does NOT affect oncogenesis but rather creates an immune response to the already established malignancy. This sentence was changed in the manuscript.

✅ Line 18: I think the statement that ICIs have “better efficacy for rapidly-proliferating cancer” compared to other treatments (chemo, radiation, and surgery) is highly controversial. Plenty of tumors proliferate fast and do not benefit/minimally benefit from ICIs to date (e.g., small cell lung cancer and pancreas).

Thank you for the comment. It is indeed a controversial statement. In our article we have cited evidence from Khan et al (2018) and Bondhopadhyay et al. (2020) that support ICIs as superior to the conventional treatment approach for a subset of tumors.

✅ Risk factors: I would also include tumor type as a risk factor: different histologies lead to different toxicity profiles. Khoja L, Day D, Wei-Wu Chen T, Siu LL, Hansen AR. Tumour- and class-specific patterns of immune-related adverse events of immune checkpoint inhibitors: a systematic review. Ann Oncol. 2017 Oct 1;28(10):2377-2385. doi: 10.1093/annonc/mdx286. PMID: 28945858.

Thank you for the comment. This paper has been cited and discussed in our manuscript as per your suggestion.

✅  Pathophysiology: The mechanisms presented exemplify how the most well-known ICIs [anti-PD(L)1 and anti-CTLA4) work. However, why do irAEs occur in the gastrointestinal tract? What is unique about GI that leads to such a high % of AEs?

This is a difficult question to answer. The presence of ICI induced adverse events does put the patient at a higher risk of other IrAEs and vice versa (Som et al, 2019). We can only postulate that this is a systemic response in our paper. We think at this stage, we do not know enough about ICI induced GIT events compared to other organs to suggest what is inherently unique about the gut. But in our paper we do cover the distinct role of the gut microbiota.

✅  Lines 159-160: Neither Stevens-Johnson nor toxic epidermal necrolysis is “commonly associated” with anti-PD(L)1 treatments. These are rare events representing less than 1% of irAEs

Thank you for pointing this out. The sentence has been changed in our manuscript.

6.1 Stomach and small bowel: I suggest the authors include a bit of the clinical presentation of such adverse events. How do these patients present? What are the typical side effects?

Thank you. This is mentioned in our manuscript. Typical symptoms include abdominal discomfort, increased stool frequencies and early satiety.

Line 223: I believe more clarity is needed in this paragraph. The authors affirm that colonoscopy has a high specificity (should be sensitivity??) in detecting mild colitis. Conversely, affirm that 37% of the clinical diagnosis of mild colitis have a normal colonoscopy, and the condition is over-diagnosed. I think these 2 passages are contradictory. In clinical practice, physicians often don´t use colonoscopy for diagnostic purposes. If the presentation is typical non-invasive tests such as CT may be performed, or even treatment may be initiated without diagnostic confirmation.  

NEW line 270: Thank you for your comment. The manuscript has been duly amended. It is true that colonoscopy is NOT the gold standard for diagnosing colitis, especially mild colitis. CT use to diagnose colitis without endoscopy finding is controversial and usually non-specific given the absence of histopathology (Hashash, 2021). What this article wants to stress is that the use of subjective symptoms to diagnose ICI colitis is unreliable and may lead to potential overdiagnosis. However, it does remain true that some initial negative biopsies later prove to be positive. This article later discusses whether a new diagnosis and grading criteria based on biopsies should be considered.

6.6: I suggest the authors include some wording about Vedolizumab. This is a potentially less immune-suppressive agent in comparison to anti-TNF alpha drugs as it is more specific to the gut.

Thank you for this suggestion. We have incorporated discussion about vedolizumab in our manuscript.

Round 2

Reviewer 2 Report

I commend the authors for the modifications. The manuscript looks stronger. 

There are no other issues from my end.